# Structure and substrate ion binding in the sodium/proton antiporter PaNhaP

David Wöhlert, Werner Kühlbrandt*, Özkan Yildiz*

Department of Structural Biology, Max Planck Institute of Biophysics, Frankfurt am Main, Germany

**Abstract** Sodium/proton antiporters maintain intracellular pH and sodium levels. Detailed structures of antiporters with bound substrate ions are essential for understanding how they work. We have resolved the substrate ion in the dimeric, electroneutral sodium/proton antiporter PaNhaP from *Pyrococcus abyssi* at 3.2 Å, and have determined its structure in two different conformations at pH 8 and pH 4. The ion is coordinated by three acidic sidechains, a water molecule, a serine and a main-chain carbonyl in the unwound stretch of trans-membrane helix 5 at the deepest point of a negatively charged cytoplasmic funnel. A second narrow polar channel may facilitate proton uptake from the cytoplasm. Transport activity of PaNhaP is cooperative at pH 6 but not at pH 5. Cooperativity is due to pH-dependent allosteric coupling of protomers through two histidines at the dimer interface. Combined with comprehensive transport studies, the structures of PaNhaP offer unique new insights into the transport mechanism of sodium/proton antiporters.

## Introduction

The Na$^+$/H$^+$ antiporter NhaP from *Pyrococcus abyssi* (PaNhaP) exchanges protons against sodium ions across the cell membrane. PaNhaP is a functional homologue of the human Na$^+$/H$^+$ exchanger NHE1, which controls intracellular pH and Na$^+$ concentration. NHE1 is an important drug target (*Karmazyn et al., 1999*), but its structure and detailed mode of action are unknown. Transport mechanisms of eukaryotic membrane proteins are conserved in the more robust prokaryotic transporters from thermophilic bacteria and archaea (*Yamashita et al., 2005*; *Boudker et al., 2007*; *Lee et al., 2013*). High-resolution structures of such homologues are of great value for understanding the mechanisms of cation/proton antiport, provided that (i) the transported substrate ions are resolved, (ii) structures of the same transporter are available in different conformations, and (iii) kinetic data of substrate binding and transport are available. In this paper we report the structure of the electroneutral Na$^+$/H$^+$ antiporter PaNhaP from the hyperthermophilic archaeon *P. abyssi* in two different conformations at pH 4 and pH 8, with the substrate ion resolved at pH 8. We show that, like NHE1, transport by PaNhaP is cooperative in a pH-dependent manner, indicating a pH-dependent allosteric interaction of protomers in the dimer.

The first structure of a cation-proton antiporter (CPA) revealed that *Escherichia coli* Na$^+$/H$^+$ NhaA (EcNhaA) is a dimer in the membrane (*Williams et al., 1999*). The 6 Å map of EcNhaA resolved 12 trans-membrane helices (TMH) in the protomer, arranged in a 6-helix bundle, plus a row of six TMHs at the dimer interface (*Williams, 2000*). A membrane dimer was also found for the NhaP1 antiporter from *Methanocaldococcus jannaschii* (MjNhaP1) (*Vinothkumar et al., 2005*; *Goswami et al., 2011*; *Paulino and Kühlbrandt, 2014*). MjNhaP1, PaNhaP and the medically important NHE1 belong to the CPA1 subfamily (*Brett et al., 2005*) of antiporters, which exchange Na$^+$ and protons with 1:1 stoichiometry and are thus electroneutral. By contrast, EcNhaA and TtNapA from *Thermus thermophilus*, as well as the eukaryotic NHA1-2 and AtChx1 (*Brett et al., 2005*), belong to the CPA2 subfamily of electrogenic antiporters, which exchange one Na$^+$ against two protons. Neither the x-ray structure of EcNhaA (*Hunte et al., 2005*) nor that of TtNapA (*Lee et al., 2013*) resolved the substrate ion.

*For correspondence: werner. kuehlbrandt@biophys.mpg.de (WK); Oezkan.Yildiz@biophys. mpg.de (ÖY)

**eLife digest** Although the membrane that surrounds a cell is effective at separating the inside of a cell from the outside environment, certain molecules must enter or leave the cell for it to work correctly. One way this transport can occur is via proteins embedded in the cell membrane, called transporters.

Transporters that are found in all organisms include the sodium/proton antiporters, which exchange protons from inside the cell with sodium ions from outside. However, exactly how the antiporter works was unknown.

Previous work suggested that the structure and activity of the sodium/proton antiporter changes as the acidity of its environment changes, but the precise details of how this occurs were unclear. Wöhlert et al. have now crystallised a sodium/proton antiporter from a single-celled organism called *Pyrococcus abyssi*, a species of archaea that has been found living in hydrothermal vents deep in the Pacific Ocean. The structures the protein takes on in different functional states were then deduced from these crystals using a technique called X-ray crystallography. Using heavy thallium ions instead of sodium ions, which are less visible to X-rays, Wöhlert et al. found the site in the antiporter where the transported ion binds as it moves through the membrane.

The antiporter has a funnel-shaped cavity that faces inwards (into the cell) in both acidic and alkaline conditions, although a second narrow channel that is open in alkaline conditions is blocked in acidic conditions by small protein rearrangements. Wöhlert et al. suggest that the differences between both structures explain how the antiporter tunes its ability to bind to the ions it transports.

Wöhlert et al. further measured the activity of the antiporter and observed that the transport of ions was most rapid under slightly acidic conditions. In more acidic conditions, the sodium ion cannot bind to the antiporter, and in an alkaline environment, the sodium ions bind too strongly to the antiporter; in both cases, the ions cannot be transported.

Comparing the findings presented here with separate work that uncovers the structure of the sodium/proton antiporter in a different species of archaea revealed very similar structures. Related transporters are also found in mammals, and defects in these transporters can lead to problems with the heart and kidneys. A better understanding of the sodium/proton antiporter structure could therefore help to develop new treatments for these conditions.

## Results

### Overall architecture of PaNhaP

Crystals of seleno-methionine derivatized PaNhaP grown at pH 8 diffracted isotropically to 3.15 Å resolution. The structure was solved by SAD (*Tables 1 and 2*). Twelve out of the 14 SeMet positions in the asymmetric unit containing one PaNhaP dimer were identified (*Figure 1—figure supplement 1*). Seen from the cytoplasm, the PaNhaP dimer is roughly rectangular, with a long axis of 90 Å and a short axis of 53 Å (*Figure 1A*, *Figure 1—figure supplement 2A*). Each protomer has 13 TMHs (H1-H13) connected by short loops or helices on the membrane surface. H4-6 and H11-13 form the 6-helix bundle, while H1-3 and H7-10 form the dimer interface. H1-6 and H8-13 are two halves of an inverted 6-helix repeat, connected by H7. Several helices are highly tilted, especially H7 and H8, which include angles of more than 45° with the membrane normal while others, in particular H6 and H10, are bent. H5 and H12 in the 6-helix bundle are discontinuous. Their cytoplasmic and extracellular halves (referred to as $H5_C$, $H5_E$ and $H12_C$, $H12_E$ respectively) are each connected by unwound stretches with antiparallel orientation, which cross one another in the centre of the protomer (*Figure 1*, *Figure 1—figure supplement 2*). The membrane surfaces are marked by three short amphipathic helices connecting H3 to H4 on the cytoplasmic side, H6 to H7 and H10 to H11 on the extracellular side. H10 protrudes by 11 Å on the cytoplasmic surface, and the helix hairpin connecting H12 to H13 protrudes by about 7 Å on the extracellular side. The loops connecting helices H1 to H2 and H8 to H9 are ~10 Å below the cytoplasmic or extracellular surface (*Figure 1B*, *Figure 1—figure supplement 2B*).

On the cytoplasmic side of the protomer, a solvent-filled ~16 Å-deep funnel, lined by H3, $H5_C$, H6, and H10, penetrates to the centre of the protomer between the 6-helix bundle and the dimer interface (*Figure 1—figure supplement 3*, *Video 1*). A second, narrow polar channel, lined by the unwound

**Table 1.** Data collection and refinement statistics

| | SeMet @ pH 8 | Thallium @ pH 8 | Native @ pH 4 |
|---|---|---|---|
| Data collection | | SLS PXII | |
| Wavelength | 0.979 | 0.979 | 0.978 |
| Space group | P2$_1$ | P2$_1$ | P6$_4$ |
| Cell dimensions | | | |
| $a$, $b$, $c$ (Å) | 54.5, 107.9, 107.9 | 54.1, 107.4, 99.8 | 109.6, 109.6, 209.6 |
| α, β, γ (°) | 90.0, 95.2, 90.0 | 90.0, 96.4, 90.0 | 90.0, 90.0, 120.0 |
| Resolution (Å) | 48.5–3.15 (3.35–3.15) | 49.6–3.20 (3.40–3.20) | 48.6–3.50 (3.72–3.50) |
| $R_{pim}$ | 0.033 (0.503) | 0.038 (0.622) | 0.021 (0.486) |
| $I / \sigma I$ | 11.9 (1.5) | 13.4 (1.8) | 19.9 (1.9) |
| CC* | 1.000 (0.943) | 1.000 (0.936) | 1.000 (0.906) |
| Completeness (%) | 99.5 (99.2) | 99.6 (99.4) | 100.0 (100.0) |
| Multiplicity | 10.8 (10.4) | 17.1 (17.4) | 9.2 (9.1) |
| Refinement | | | |
| Resolution (Å) | 48.5–3.15 (3.35–3.15) | 49.6–3.20 (3.40–3.20) | 48.6–3.5 (3.72–3.5) |
| Unique reflections | 38,952 | 34,763 | 33,232 |
| Reflections in test set | 2111 | 1884 | 1782 |
| $R_{work}$/$R_{free}$ (%) | 23.8/27.8 (31.8/39.9) | 24.8/29.5 (35.9/43.4) | 24.1/26.4 (31.8/35.6) |
| CC(work)/CC(free) | 0.843/0.898 (0.842/0.760) | 0.861/0.754 (0.813/0.713) | 0.791/0.935 (0.749/0.617) |
| Wilson B-Factor (Å²) | 133 | 81 | 146 |
| No. atoms in AU | 6715 | 6651 | 6592 |
| Protein | 6582 | 6560 | 6560 |
| Ligands | 129 | 81 | 31 |
| Water | 4 | 10 | 1 |
| r.m.s. deviations: | | | |
| Bond lengths (Å) | 0.003 | 0.003 | 0.009 |
| Bond angles (°) | 0.758 | 0.714 | 1.002 |

stretches of H5$_C$, H12$_C$ and the cytoplasmic halves of H6 and H13, extends from the cytoplasmic surface to the region near the deepest point of the funnel (*Figure 1—figure supplement 3*, *Video 1*). On the extracellular side, a deep cavity on the twofold axis of the dimer, lined by interface helices H1, H3, H8 and H10, extends ~27 Å into the hydrophobic protein interior. Electron density in this cavity indicated bound lipid (*Figure 2*), which was identified by thin-layer chromatography as phosphatidyl ethanolamine (PE), carried over from the *E. coli* expression host. The lipid stretches from the 6-helix bundle of one protomer to the interface helices H3 and H10 of the other, providing a hydrophobic link between them. The cavity is large enough to accommodate two lipids, only one of which was resolved in the dimer (*Figure 2*). The surface potential of the dimer indicates clusters of charged residues on both sides of the membrane (*Figure 2—figure supplement 1*). The cytoplasmic ends of H10 and H7 are positively charged, carrying a total of seven lysine and arginine residues. The deep funnel on the cytoplasmic surface is lined by negative charges, which would attract positively charged substrate ions.

## The ion-binding site

Crystals of PaNhaP grown at pH 8 soaked with thallium acetate diffracted to 3.2 Å (*Table 1*). Two thallium ions were identified in the dimer by anomalous scattering, one each near the deepest point of the cytoplasmic funnel in the two protomers (*Figure 1—figure supplement 3*, *Video 1*, *Figure 3A*). The Tl$^+$ ions were located ~14 Å below the cytoplasmic surface and ~22 Å from the extracellular surface. The ion-binding site is accessible from the cytoplasm but not from the extracellular side, so that the structure shows the inward-open conformation of PaNhaP (*Figure 1—figure supplement 3*, *Video 1*).

**Table 2.** Data collection and phasing statistics

| | Dataset 1 | Dataset 2 | Merge |
|---|---|---|---|
| Data collection | SLS PXII | | |
| Wavelength | 0.979 | 0.979 | 0.979 |
| Space group | P2$_1$ | P2$_1$ | P2$_1$ |
| Cell dimensions | | | |
| $a$, $b$, $c$ (Å) | 54.7, 109.0, 110.8 | 54.6, 108.3, 110.5 | 54.7, 108.9, 110.7 |
| α, β, γ (°) | 90.0, 94.6, 90.0 | 90.0, 95.0, 90.0 | 90.0, 94.7, 90.0 |
| Resolution (Å) | 49.3–3.8 (3.97–3.8) | 49.0–3.8 (3.97–3.8) | 49.2–3.8 (3.97–3.8) |
| $R_{pim}$ | 0.029 (0.470) | 0.034 (0.228) | 0.036 (0.315) |
| $I / \sigma I$ | 13.8 (2.1) | 12.4 (3.9) | 14.5 (2.9) |
| CC* | 1.000 (0.929) | 0.996 (0.985) | 1.000 (0.981) |
| Completeness (%) | 99.7 (99.7) | 99.7 (99.6) | 100 (100) |
| Multiplicity | 24.5 (16.5) | 9.1 (9.5) | 33.0 (25.8) |
| Phasing | | | |
| CCanom | | | 0.348 |
| Anom slope | | | 1.061 |
| FOM after Phasing (Refmac) | | | 0.230 |
| FOM after DM (Parrot) | | | 0.594 |

Like Na$^+$ and Li$^+$, but unlike K$^+$, Tl$^+$ is a substrate of PaNhaP (*Figure 4*). The thallium ions and their surroundings provide a unique view of the ion-binding site and substrate ion coordination in sodium-proton antiporters (*Figure 3B,C*). Three acidic side chains in three different TMHs contribute to substrate ion-binding. The carboxyl groups of Glu73 in H3 and Asp159 in H6 coordinate the substrate ion directly. Asp130 in the unwound stretch of H5 interacts with the ion via a bound water molecule (*Figure 3*). The main-chain carbonyl of Thr129, likewise in the unwound stretch of H5, and the hydroxyl side chain of Ser155 in H6 provide two additional ligands, bringing the total up to five. The ion coordination geometry is that of a distorted trigonal bipyramid, with Asp159, Ser155 and the water molecule forming a triangle around the central substrate ion, and the Thr129 main chain carbonyl and Glu73 at the tips of the bipyramid (*Figure 3B,C*).

The second, narrow polar channel next to the cytoplasmic funnel (*Figure 1—figure supplement 3*, *Video 1*) leads to an enclosed polar cavity near Asp93, Thr129, Asn158 and the ion pair Glu154/Arg337, which are highly conserved in the CPA1 antiporters (*Goswami et al., 2011*). A water molecule in the enclosed cavity links the functionally important groups that surround it. The Glu154/Arg337 ion bridge and Thr129 separate the cavity from the narrow polar channel. The ion-binding site at the end of the cytoplasmic funnel is accessible from both the polar cavity and the narrow polar channel via Thr129, Ser155, and Asn158 (*Figure 1—figure supplement 3*, *Video 1*, *Figure 3*).

## Conformational changes at pH 4

The structure of PaNhaP crystals grown at pH 4 was determined at 3.5 Å (*Table 1*) by molecular replacement. As at pH 8, the ion-binding site is accessible from the cytoplasm via the cytoplasmic funnel, but not from the extracellular side. Both structures therefore show an inward-open state. In contrast to the pH 8 structure, the second narrow polar channel is blocked at pH 4 by rearrangements of the surrounding residues Ile151, Phe355, Gly359. The most conspicuous differences to the pH 8 structure are observed near the dimer interface. At pH 8, the His292 sidechains in H10 of the two protomers form a 15 Å chain of hydrogen bonds with the Glu233 residues near the cytoplasmic ends of H8 (*Figure 5*, *Video 2*, *Figure 5—figure supplement 1*). At pH 4, each of the two histidines moves by 6–8 Å, apparently due to electrostatic repulsion upon protonation at acidic pH (*Figure 2—figure supplement 1*). This pH-induced conformational change disrupts the chain of hydrogen bonds linking the two protomers (*Figure 5—figure supplement 1*). Other major pH-induced changes are found in

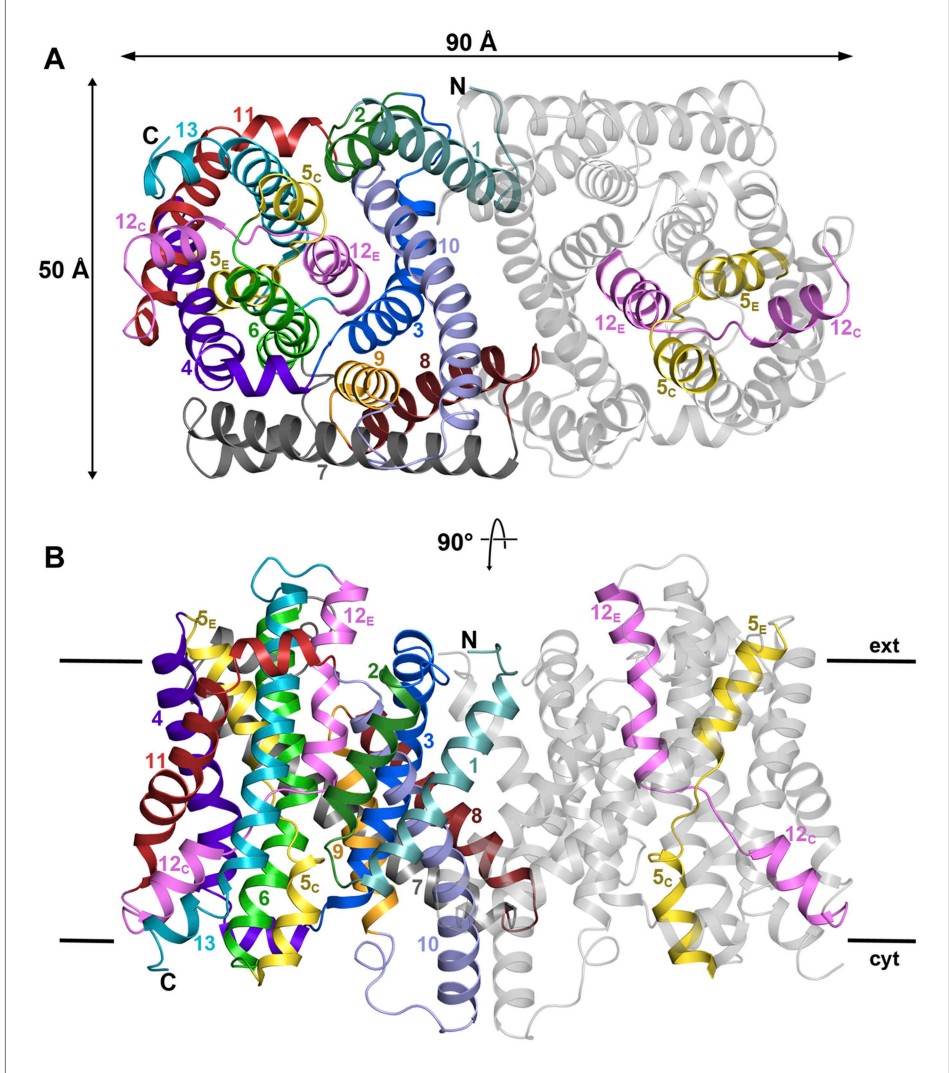

**Figure 1**. PaNhaP at pH 8. (**A**) Cytoplasmic view of the PaNhaP dimer. Helices H1 to H13 are color-coded and numbered in one protomer. In the other protomer only the partly unwound helices H5 and H12 are coloured. (**B**) Side view with the C-terminus of helix H13 on the cytoplasmic side.

The following figure supplements are available for figure 1:

**Figure supplement 1**. Experimental electron density map of PaNhaP.

**Figure supplement 2**. X-ray structure of PaNhaP.

**Figure supplement 3**. Hydrophilic cavities in PaNhaP.

the ion bridges linking the protomers across the dimer interface (*Figure 5—figure supplement 1*). At pH 8, Arg25/Glu228 and Arg26/Asp231 connect the cytoplasmic ends of H8 and H1, while the Glu8/Arg249 bridge links the extracellular ends of these helices. At pH 4, all six ion pairs break, apparently due to partial protonation of the acidic sidechains, so that each protomer tilts away from the dimer interface (*Video 2*, *Figure 5—figure supplement 1*).

Other significant pH-induced differences occur at the N-terminus of the protomer, where residues 3–6 become ordered at pH 4, so that H1 extends by one turn, and shifts by 3 Å towards the cytoplasmic side (*Figure 5—figure supplement 1*). In the ion-binding site itself, the sidechain of Asp130 in protomer A moves by 2.7 Å into the space that is occupied by the substrate ion at pH 8

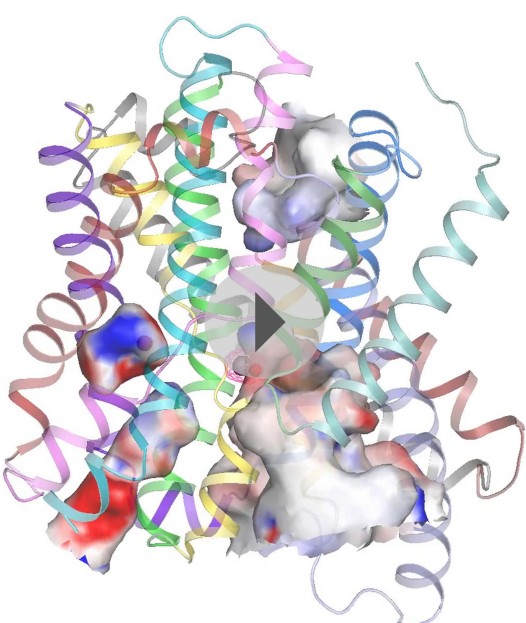

**Video 1**. Movie of PaNhaP monomer with hydrophilic cavities.

(*Figure 5—figure supplement 2A*). This movement, which is not observed in the other protomer (*Figure 5—figure supplement 2B*), could displace a bound substrate ion or prevent ion binding. At pH 4, the conserved Asn158 that interacts with Asp93 at pH 8 moves by ~2.5 Å towards the ion-coordinating Asp159 in protomer A, forming a H-bond network with Thr129 and the main-chain carbonyls of Glu154 and Ser155. In this way, the reorientation of Asn158 may regulate access to the ion-binding site through the narrow polar channel (*Figure 1—figure supplement 3*, *Video 1*).

A chain of hydrogen bonds stretches from Glu290 in H10 via His75 near the cytoplasmic end of H3 to Glu73, the only ion-coordinating side-chain from one of the interface helices (*Figure 3A*). This residue most likely relays allosteric changes from the dimer interface to the ion-binding site. An opening of the ion bridges that link H1 and H8 and the movement of the adjacent H2 in the pH 8 to pH 4 transition is likely to affect substrate binding via Tyr31, which is within H-bonding distance of the substrate-coordinating Asp130 (*Figure 5—figure supplement 2*). In this way, the conformational changes caused by repulsion of the protonated histidines 292 at the dimer interface are relayed to the ion-binding site to modulate the $Na^+$ binding affinity in a pH-dependent manner (*Figure 6*).

## pH-dependent cooperativity

$^{22}Na$ uptake into reconstituted PaNhaP proteoliposomes is strongly pH-dependent (*Figure 6A*). Transport activity was highest at pH 5, dropping to about 75 % at pH 6, 20 % at pH 7, and to background level at pH 8. At pH 4, the activity was about 5 % of the peak value at pH 5, resulting in a roughly bell-shaped pH profile. Sodium uptake measurements performed with reconstituted, inside-acidic proteoliposomes (*Figure 6B,C*) or sodium efflux measurements under symmetrical pH conditions (*Figure 6—figure supplement 1A*) showed comparable transport behaviour at basic pH. Valinomycin had no effect on the transport rate (*Figure 6—figure supplement 1B*), demonstrating that PaNhaP is electroneutral. Measurements of $^{22}Na^+$ uptake at pH 6 revealed clear positive cooperativity, with a Hill coefficient of 1.9 (*Figure 6C*, *Figure 6—figure supplement 2B*). Since PaNhaP forms stable dimers in detergent solution and each protomer binds only one substrate ion at a time, this indicates that the interaction of protomers across the dimer interface is allosteric, such that at pH 6, an ion binding to one protomer increases the binding affinity of the other, as indicated by the $K_{0.5}$ value of 25 μM (*Figure 6C*), compared to the $K_m$ of 506 μM at pH 5 (*Figure 6C*). At the pH 5 activity maximum the Hill coefficient was ~1, indicating non-cooperative transport (*Figure 6B*, *Figure 6—figure supplement 2*). Note that the pH-dependent allosteric change of the dimer is different from the inside-open to outside-open transition in the transport cycle of the protomer.

## Transport activity

At room temperature, $v_{max}$ of PaNhaP at the pH 5 activity maximum was 87.9 nmol · $min^{-1}$ · $mg^{-1}$, giving a transport rate of 4.4 ± 0.4 $Na^+$ ions per minute for each protomer. Between 20°C and 45°C, $v_{max}$ grew exponentially by a factor of 2.1 for every 5°C rise in temperature (*Figure 7A,B*) according to the Arrhenius equation. Extrapolation to 100°C, the physiological temperature for *P. abyssi*, suggests a rate of about 5000 ions per second. Note that temperature affects the transport rate but not substrate binding (*Figure 7C,D*).

Residues involved in substrate binding were replaced and the transport activity of mutant proteins was measured in proteoliposomes. Replacement of both ion-coordinating aspartates (Asp130 and Asp159) by serine abolished transport completely (*Figure 8A*), whereas mutation of Glu73 to alanine

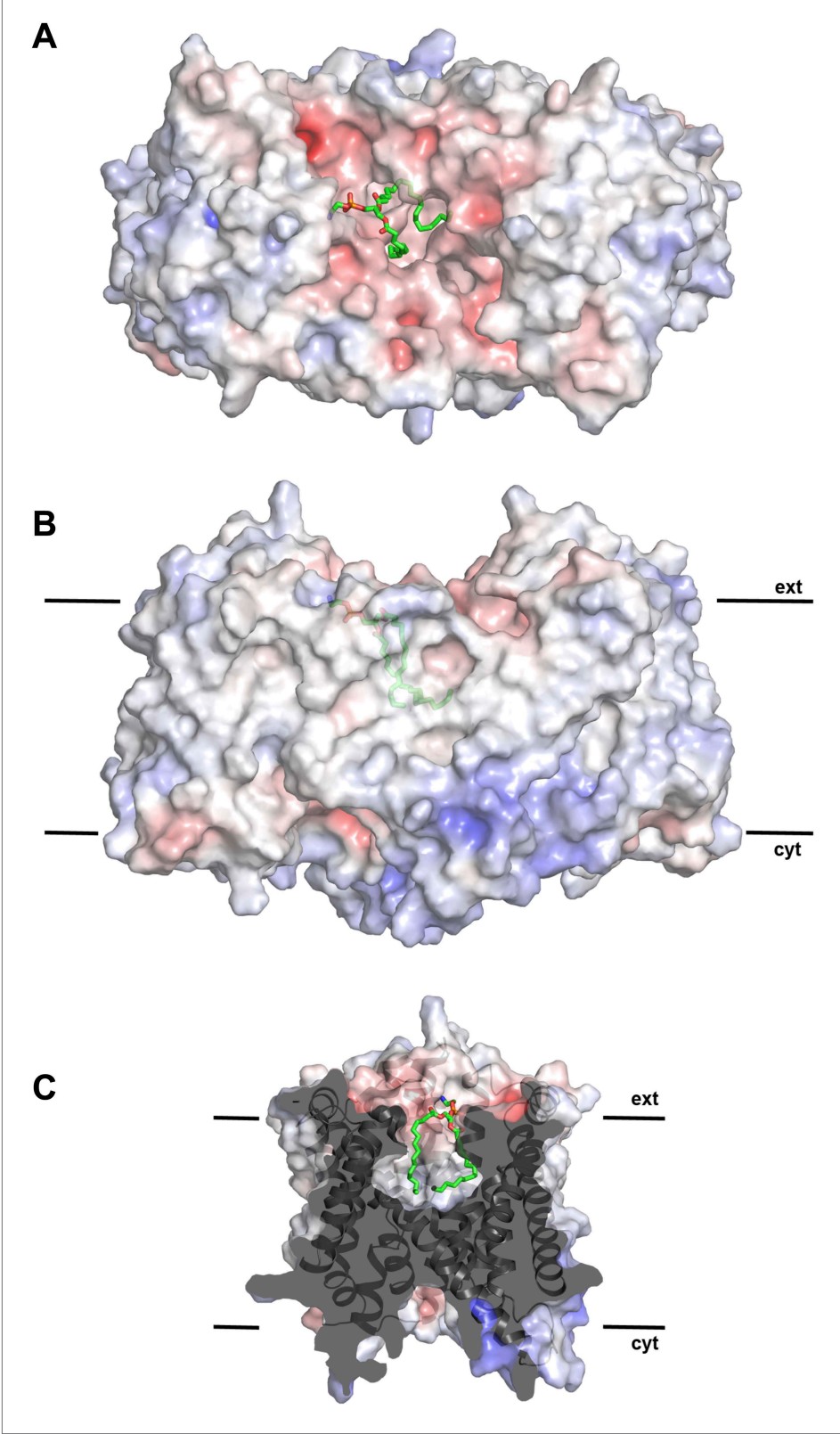

**Figure 2**. Hydrophobic extracellular cavity with bound lipid. (**A**) One lipid molecule (PE, green) in the cavity between the two protomers in the dimer contributes to the hydrophobic contacts across the dimer interface. The extracellular surface is slightly negatively charged. (**B**) The alkyl chain of the lipid extends to the center of the
*Figure 2. Continued on next page*

*Figure 2. Continued*
molecule. (**C**) The lipid-facing surface of the central cavity is mainly hydrophobic. The surface potential was calculated at pH 7.0 by APBS.
The following figure supplement is available for figure 2:

**Figure supplement 1**. pH-dependent charge distribution.

increased the activity (*Figure 8B*), most likely because the substrate ion is released more readily from the binding site. Changing Ser155 to alanine had no significant effect, but mutation of Thr129 to valine that takes this position in eukaryotic CPA1 transporters (*Goswami et al., 2011*), reduced the activity significantly. This was surprising, because Thr129 coordinates the substrate ion not by its sidechain but by its main-chain carbonyl. However the Thr129 sidechain is a potential interaction partner of the conserved Asn158 that may control access to the ion-binding site through the narrow polar channel. A hydrophobic valine in place of Thr129 would interrupt the local network of hydrogen bonds, which could affect ion binding or proton translocation. A mutant in which His292 was replaced by cysteine migrates as a dimer under oxidizing conditions in SDS-PAGE (*Figure 8—figure supplement 1A*). The activity of the crosslinked dimer was 35 % of wildtype (*Figure 8—figure supplement 1B*). Under reducing conditions, when the disulfide bridge between the protomers is broken, activity increases to 150 % of wildtype, highlighting the importance of this position for the regulation of transport.

## Discussion

### Ion coordination

The trigonal bipyramidal coordination geometry of sodium ions observed in PaNhaP is not uncommon in membrane transporters (*Penmatsa et al., 2013*). The same geometry is found in c-rings of $Na^+$-translocating F-type ATPase of *I. tartaricus* and *F. nucleatum* (*Meier et al., 2009*; *Schulz et al., 2013*), which, like the archaeal CPA1 antiporters, bind and release $Na^+$ in rapid exchange. Although the ion radius of monovalent $Tl^+$ (1.5 Å) is similar to that of $K^+$ (1.44 Å) and larger than that of $Na^+$ (1.12 Å) (*Shannon, 1976*; *Cotton and Wilkinson, 1988*), $Tl^+$ is able to replace $Na^+$ in PaNhaP. The same has been found for the $Na^+$-dependent aspartate transporter $Glt_{Ph}$ (*Boudker et al., 2007*), the mammalian glutamate transporter EAAC1 (*Tao et al., 2008*) and fructose-1,6-biphosphatase (*Villeret et al., 1995*). In $Glt_{Ph}$, $Na^+$ but not $K^+$ competes for $Tl^+$ binding, and $Tl^+$ inhibits $Na^+$-driven aspartate transport (*Boudker et al., 2007*). Coordination geometry and ligand distances for $Tl^+$ in PaNhaP are similar to those typically found for protein-bound $Na^+$ in the PDB (*Harding, 2002*). The larger ion radius of $Tl^+$ may account for the lower transport rate in PaNhaP. However, $Tl^+$ is a much better substrate than $K^+$, which is not transported at all (*Figure 4*). The selectivity for $Na^+$ over $K^+$ is reminiscent of the striking selectivity of sodium channels, which is thought to be related to ion solvation (*Roux et al., 2011*). Presumably, the same principle applies to the $Na^+/H^+$ antiporters. The water molecule between the sidechain of Asp130 and $Tl^+$ indicates that the bound substrate ion retains part of its hydration shell, as complete dehydration is energetically unfavourable.

In PaNhaP, all ion-binding residues are found in the first half of the inverted repeat. Interestingly, the structure and interaction of the ion-coordinating Asp159 and Ser155 in H6 resemble those of the inversely oriented Glu408 and Ser404 in H13 in the second half of the inverted repeat (*Figure 3A,D*). This may imply that an early form of the CPA1 antiporters, which must have arisen by gene duplication of an unknown precursor, had a second, symmetrical ion-binding site that has been lost in the course of evolution. Arg362 in the unwound stretch of H12, which is essential in MjNhaP1 (*Hellmer et al., 2003*) and completely conserved in the CPA1 antiporters (*Hellmer et al., 2003*; *Goswami et al., 2011*), may be a tethered positive charge that takes the place of the $Na^+$ ion in the second half of the inverted repeat, in a way similar to the arginine that replaces the co-transported $Na^+$ in the sodium-independent substrate/product antiporter CaiT (*Kalayil et al., 2013*).

### Regulation of transport activity

The transport activity of PaNhaP is highest at pH 5 and declines at higher or lower pH. The resulting bell-shaped pH profile is explained in terms of the $Na^+$ affinity of the acidic residues in the substrate-binding pocket. The protonation state of these is likely to affect the affinity of the binding site

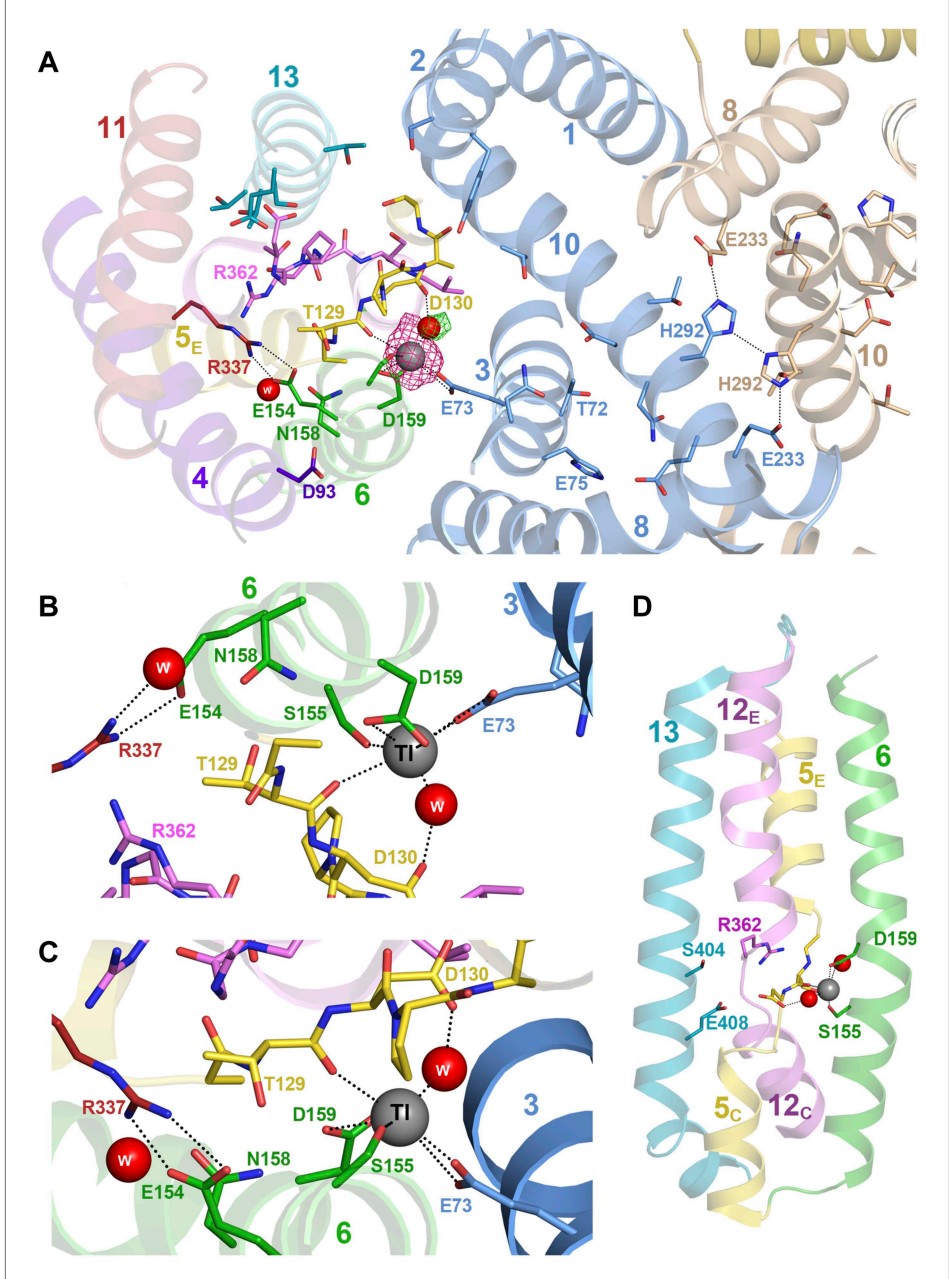

**Figure 3**. Substrate ion coordination in PaNhaP. (**A**) Section view of the ion-binding site and interface region of PaNhaP from the cytoplasmic side. Interface helices of the two protomers are shown in blue and beige, respectively. The acidic side chains of Glu73, Asp159, a water molecule held by Asp130, the hydroxyl group of Ser155 and the main-chain carbonyl of Thr129 coordinate the substrate ion. The anomalous density for the $Tl^+$ ion (grey sphere) in the substrate-binding site between helix H3, H6 and the unwound stretch of H5 is shown in magenta at $4\sigma$. The $3\sigma$ omit map for the $H_2O$ molecule next to $Tl^+$ is green. The water molecule near Glu154 and Asn158 is not directly involved in ion coordination. (**B**, **C**) Detailed views of the substrate-coordinating residues from the extracellular and cytoplasmic side, respectively. (**D**) Side view of core helices and substrate-binding residues in the 6-helix bundle.

for $Na^+$. At low pH, most if not all of the ion-coordinating carboxyl sidechains (Glu73, Asp130, and Asp159) would be protonated, resulting in reduced affinity for $Na^+$, as has been shown for MjNhaP1 by electrophysiological measurements on solid-supported membranes (**Calinescu et al., 2014**). At pH 5–7 these carboxyl sidechains would be increasingly deprotonated and able to bind and release $Na^+$ ions, as is necessary for transport. At pH > 7, the ion-binding site is predominantly deprotonated

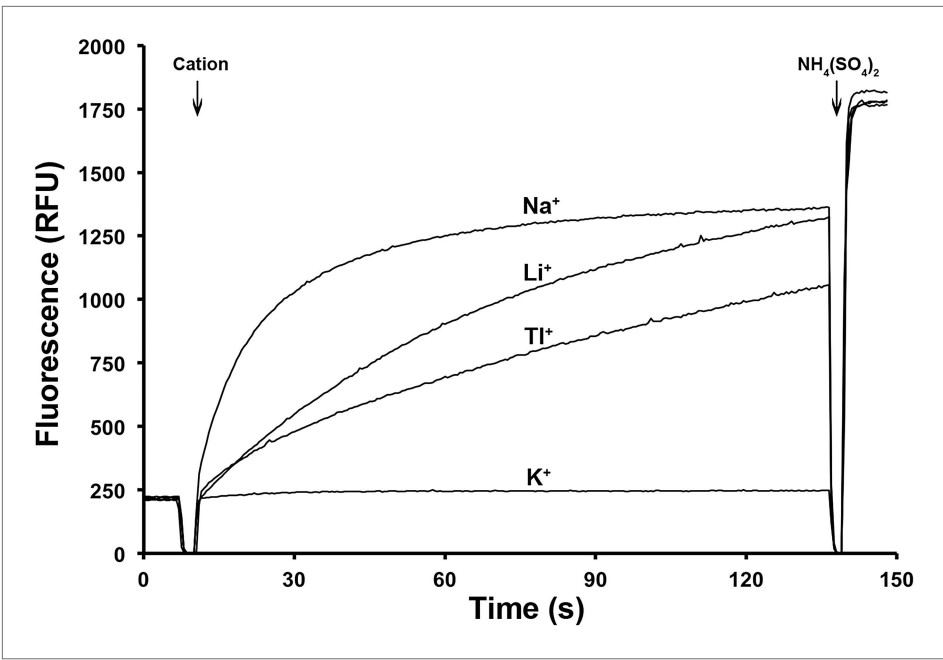

**Figure 4**. Ion selectivity of PaNhaP. Ion selectivity was determined by acridine orange fluorescence at pH 6. $Na^+$, $Li^+$, $Tl^+$ are transported by PaNhaP, $K^+$ is not.

and negatively charged (*Figure 2—figure supplement 1*), resulting in an increased $Na^+$ affinity. As a result, the transport rate would decrease, as the ions are bound more tightly. This is consistent with the increased transport rate of the E73A mutant, which has one less carboxyl in the binding site, hence releases $Na^+$ more easily (*Figure 8B*). In addition, the propagation of the pH-induced conformational changes at the dimer interface via Glu73 or Tyr31 would modulate the binding site by changing the coordination geometry for the ions (*Figure 5—figure supplement 1A*, *Figure 5—figure supplement 2*). Future structure-based molecular dynamics simulations should show how the protonation state of each of these residues influence the affinity of the binding site for $Na^+$ in a pH-dependent manner.

The pH-dependent transport activity of PaNhaP suggests a self-regulatory mechanism for the binding site rather than regulation by a separate pH sensor as proposed for EcNhaA (*Herz et al., 2010*; *Diab et al., 2011*; *Schushan et al., 2012*). At the pH 5 activity maximum of PaNhaP, transport is not limited by $Na^+$ affinity. Under these conditions, substrate binding of the PaNhaP dimer is noncooperative, but unexpectedly it becomes cooperative at pH 6. Cooperative ion binding is most likely mediated by Glu73 and may be important for controlling the intracellular pH at neutral or basic pH, where a cooperative increase in $Na^+$ affinity would gradually inhibit substrate release and slow down transport. This may be a safety mechanism to protect the organism against excessive influx of $Na^+$, and hence efflux of protons, at rising pH, which may be critical for survival.

The medically relevant but elusive human $Na^+/H^+$ exchanger NHE1 is a dimer (*Fafournoux et al., 1994*) like PaNhaP. Several other common features, including high sequence homology (*Goswami et al., 2011*) especially of the unwound stretches in H5 and H12, key residues in the ion binding site such as Ser155, Asp130 and the ND motif, the functionally important Arg337 and Arg362 (*Hellmer et al., 2003*), as well as pH profiles and transport kinetics suggest that the archaeal and mammalian CPA1 antiporters (*Fuster et al., 2008*) work essentially in the same way. Remarkably, NHE1 also shows pH-dependent $Na^+$ cooperativity, with a Hill coefficient of 1.8 at pH 6.8 that drops to ~1 at pH 6 (*Fuster et al., 2008*). The PaNhaP structure thus serves as an excellent model for the membrane part of NHE1. Molecular details of allosteric regulation in NHE1 are likely to be different, as the His292 that reorients in response to pH in PaNhaP is not conserved (*Goswami et al., 2011*).

The electrogenic CPA2 antiporters, such as EcNhaA or TtNapA, which exchange two protons against one $Na^+$, have two conserved aspartates in place of the ND motif in H6. In terms of its overall

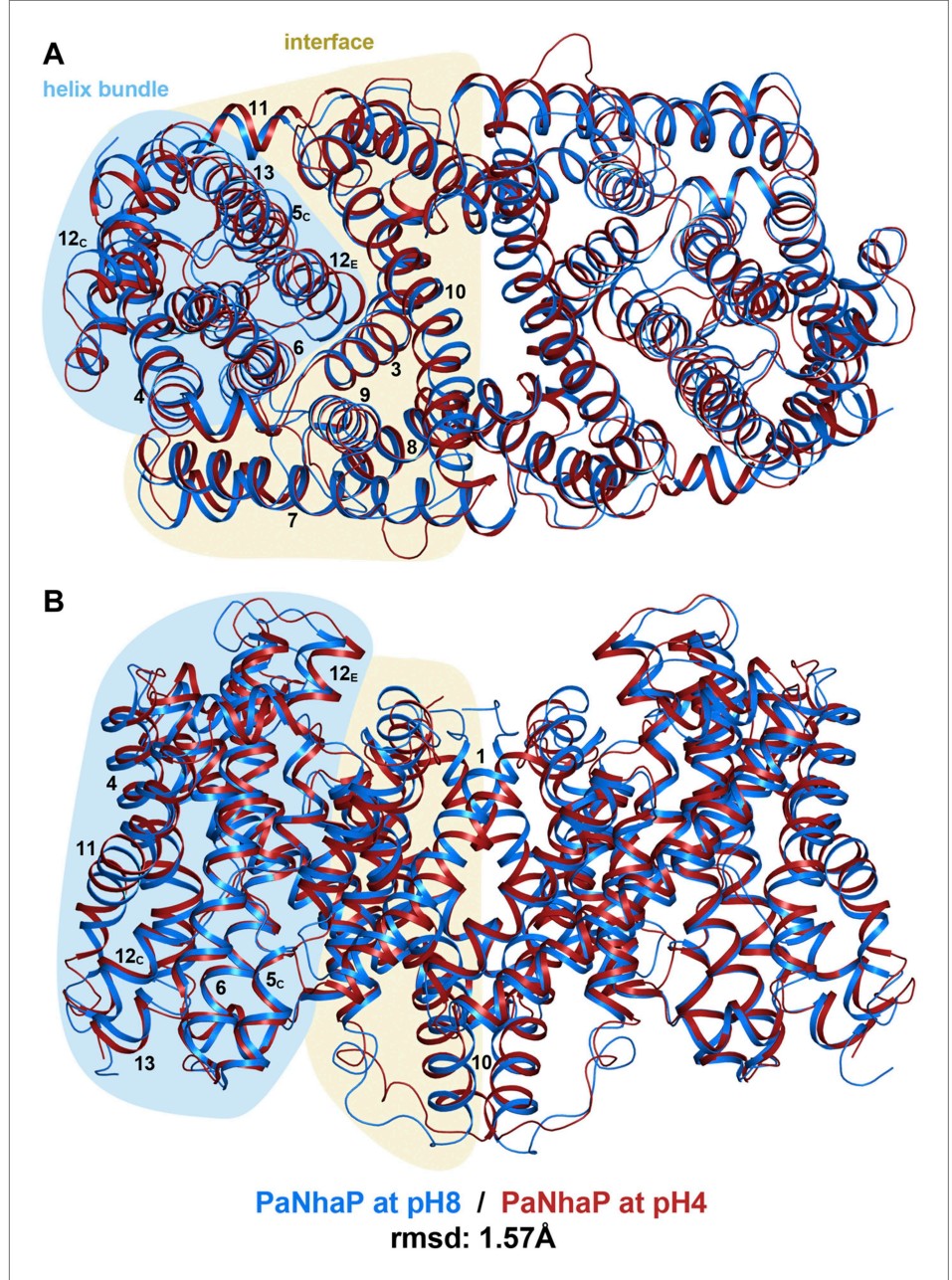

**PaNhaP at pH8** / **PaNhaP at pH4**
**rmsd: 1.57Å**

**Figure 5**. pH-induced conformational changes in the PaNhaP dimer. (**A**) cytoplasmic view, (**B**) side view as in **Figure 1**. At pH 4 (red), helix H4 moves towards the cytoplasm by 1.5 Å. Within the 6-helix bundle, the extracellular ends of helix $H5_E$ and H6 move towards H12 by ~1.5 Å. Helix H11 and H13 tilt by about 2–3° each, such that the cytoplasmic end of helix H11 moves towards $H12_C$, which shifts by a similar amount in the same direction. The extracellular end of helix $H12_E$ moves towards helix H3 by ~3 Å. The rmsd between the structures at pH 8 and pH 4 is 1.57 Å.

The following figure supplements are available for figure 5:

**Figure supplement 1**. pH-induced conformational changes at the dimer interface.

**Figure supplement 2**. pH-induced conformational changes in the substrate binding site.

structure, TtNapA (**Lee et al., 2013**) is more similar to PaNhaP than to EcNhaA (**Hunte et al., 2005**), especially with respect to the dimer interface. The tertiary structure of the CPA antiporters is thus not a diagnostic of electroneutral or electrogenic transport.

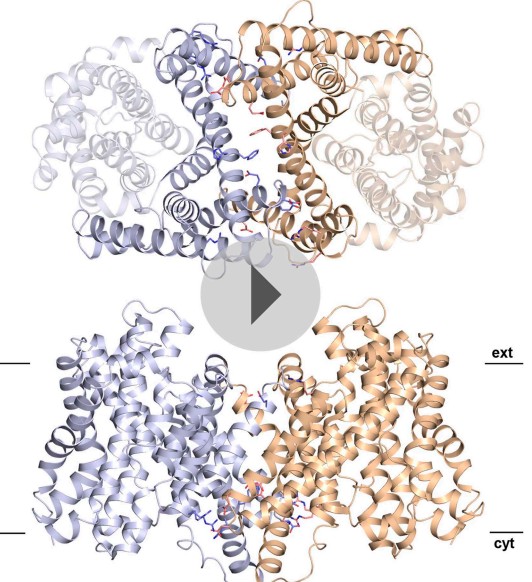

**Video 2**. pH-induced conformational changes in PaNhaP. A morph between the pH 4 and pH 8 structures reveals only small changes in the 6-helix bundle, but significant rearrangements at the dimer interface. Six ion bridges that lock the two protomers together at pH 8 break at pH 4. As a result, the two protomers tilt away from each other at lower pH. His292 has a pivotal role in the allosteric pH-dependent protomer interaction. At pH 4, the protonated His292 side chains on the cytoplasmic side of the dimer interface repel one another by electrostatic repulsion, resulting in a ~7 Å movement that disrupts the hydrogen-bonding network with Glu233.

## Mechanisms of ion binding and release

In *Pyrococcus*, the Na$^+$ gradient required for ATP synthesis is maintained by specific antiporters (*McTernan et al., 2014*). We therefore assume that PaNhaP, like human NHE1, utilizes the Na$^+$ gradient across the membrane (*Cohen et al., 2003*) for pH homeostasis. Protons, probably in the form of hydronium ions (H$_3$O$^+$), can reach the binding pocket either through the cytoplasmic funnel or through the narrow polar channel (*Figure 9*). Only small rearrangements of the residues lining this channel would be required for H$_3$O$^+$ to pass. Using the second narrow polar channel for proton translocation would physically separate the routes for Na$^+$ and H$_3$O$^+$ on the cytoplasmic side, which may be an advantage as the two ion currents flow in opposite directions. It would also explain why residues that line this channel, in particular the Glu154/Arg337 ion bridge and Asn158, which do not participate in ion coordination, are so highly conserved in the family. Molecular dynamics simulations and functional analysis of suitable mutants will be required to differentiate between the two proton paths, which both appear equally likely on the basis of the x-ray structures.

## Materials and methods

### Cloning, expression and purification

A codon-optimized synthetic gene for the Na$^+$/H$^+$ antiporter from *Pyrococcus abyssi* (WP_010868413.1) was cloned into a vector with a C-terminal cysteine protease domain fusion as described previously for soluble proteins (*Shen et al., 2009*). Mutations were introduced by site-directed mutagenesis (*Braman et al., 1996*). The resulting plasmids were used to transform *E. coli* C41-(DE3) cells. The protein was expressed for 10 hr at 37°C in ZYM-5052 autoinduction medium (*Studier, 2005*).

Membranes were isolated from a 12 l culture and resuspended at 15 mg/ml protein in 20 mM Tris pH 7.4, 250 mM sucrose, 140 mM choline chloride. The suspension was diluted 1:3 in solubilization buffer (20 mM Tris pH 7.4, 150 mM NaCl, 30 % Glycerol and 1.5 % Cymal-5). After solubilization overnight at 4°C the solution was clarified by centrifugation at 100,000×*g* for 1 hr. The supernatant was supplemented with 5 mM imidazole, incubated for 2 hr with Talon resin (Clontech, Mountain View, CA) at 4°C and loaded on a Biorad column. Unspecifically bound proteins were eluted with washing buffer (20 mM Tris pH 7.4, 300 mM NaCl, 10 mM imidazole and 0.15 % Cymal-5). PaNhaP was cleaved off the column by incubating the beads in elution buffer (20 mM Tris pH 7.4, 300 mM NaCl, 0.15 % Cymal-5, 20 µM inositol-hexaphosphate) for 30 min. The eluted protein was concentrated to 5 mg/ml using a concentrator with 50 kDa cutoff and applied to a Superdex 200 size exclusion column equilibrated with 10 mM Na-Citrate pH 4.0, 300 mM NaCl and 0.15 % Cymal-5. Antiporter-containing fractions were pooled and concentrated to 5 mg/ml. The concentrated protein solution was diluted 1:4 with the same buffer containing 100 mM NaCl and re-concentrated as above. Selenomethionine (SeMet) labeled protein was expressed in PASM-5052 autoinduction medium (*Studier, 2005*) and purified as described for the native protein in the presence of 5 mM β-mercaptoethanol throughout all purification steps. β-mercaptoethanol was exchanged to 1 mM TCEP (Tris-(2-carboxyethyl) phosphine) in the final size-exclusion chromatography step.

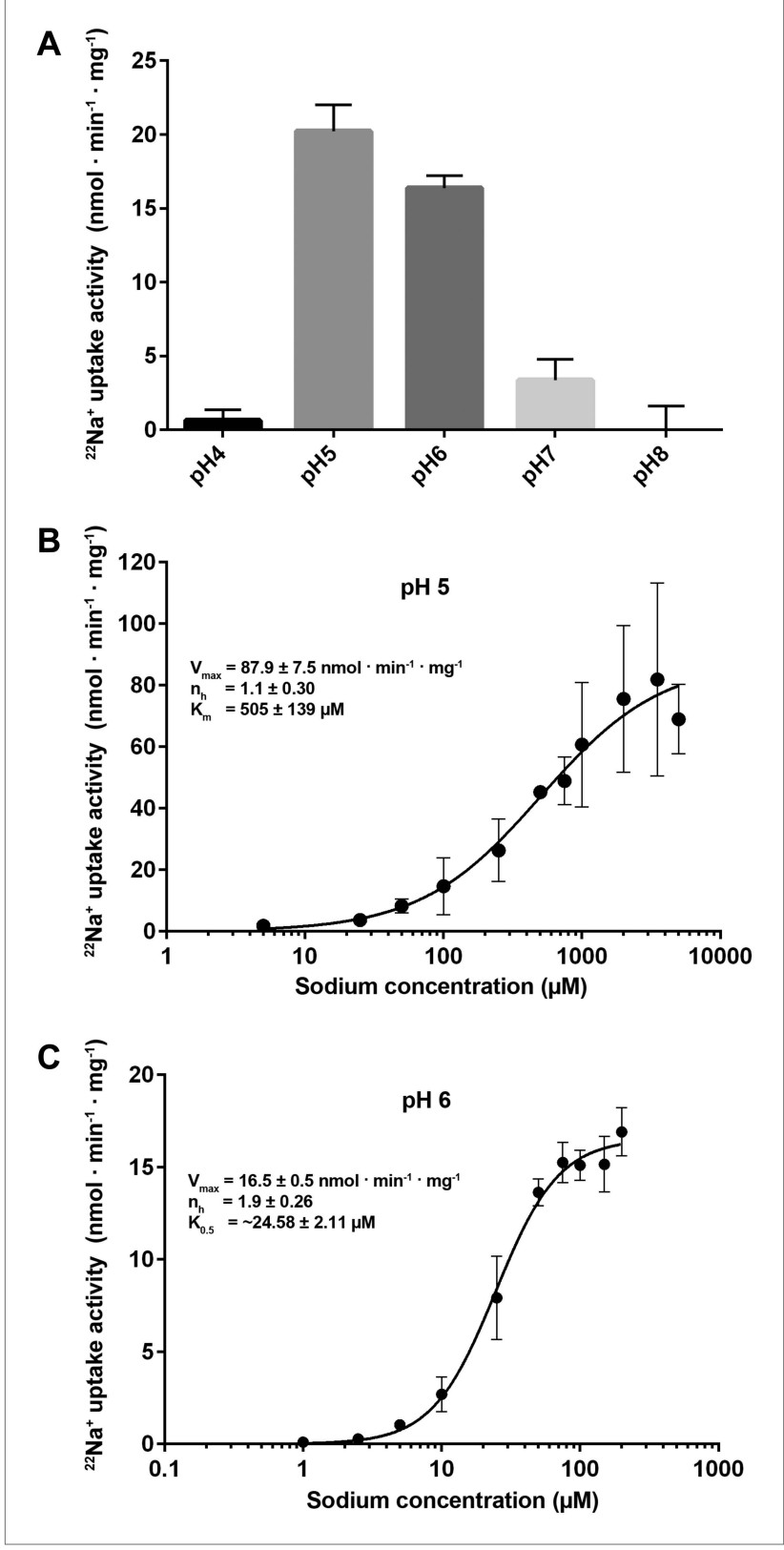

**Figure 6**. Transport activity of PaNhaP. (**A**) pH dependence of transport activity determined by $^{22}Na^+$ uptake with inside-acidic PaNhaP proteoliposomes. The antiporter is active at pH 5 and pH 6; at pH 4 and pH 7 transport drops to background level. (**B**) Concentration-dependent $^{22}Na^+$-uptake by inside-acidic PaNhaP proteoliposomes at pH 5

*Figure 6. Continued on next page*

*Figure 6. Continued*

gives a $v_{max}$ of 87.9 ± 7.5 nmol · $min^{-1}$ · $mg^{-1}$ at room temperature, indicating a transport rate of 4.4 $Na^+$ ions per protomer per minute. At pH 5 the Hill coefficient ($n_h$) is 1.1 ± 0.30, indicating non-cooperative transport. (**C**) At pH 6, transport is cooperative, with a Hill coefficient of 1.9 ± 0.26, indicating allosteric coupling of the two ion-binding sites in the dimer. $v_{max}$ at room temperature decreases to 16.5 ± 0.5 nmol · $min^{-1}$ · $mg^{-1}$.

The following figure supplements are available for figure 6:

**Figure supplement 1**. Sodium efflux measurements.

**Figure supplement 2**. Eadie-Hofstee plots.

## Reconstitution

*E. coli* polar lipids (EPL, Avanti Polar Lipids, Inc., Alabaster, AL) were dried under nitrogen and resuspended in reconstitution buffer. Unilamellar vesicles were prepared by extruding the resuspended lipids using an extruder (Avestin, Inc., Canada) with 400 nm polycarbonate filters. Vesicles were destabilized by stepwise addition of n-octyl-β-D-glucoside (OG). The process was monitored at 540 nm. Addition of OG was stopped at around 1 % final concentration when the absorbance decreased rapidly. Protein was added to the destabilized liposomes at a lipid-to-protein ratio (LPR) of 100:1 and incubated for 1 hr at room temperature. The solution was dialyzed (14 kDa cutoff) overnight at room temperature against detergent-free reconstitution buffer. Biobeads (SM2, Biorad, Hercules, CA) were added to the dialysis buffer to ensure complete removal of the detergent. Proteoliposomes were centrifuged at 300,000×*g* for 20 min and washed once with reconstitution buffer. Washed proteoliposomes were pelleted again and resuspended at ~60 mg/ml lipid in reconstitution buffer for further use.

## Fluorescence assays

PaNhaP was reconstituted into proteoliposomes in 10 mM choline citrate/Tris pH 6–8, 200 mM NaCl and 5 mM KCl. To start the reaction 2 µl of proteoliposome suspension were diluted into 2 ml reaction buffer (10 mM choline-citrate/Tris at same pH, 5 mM KCl, 2 µM acridine orange). Emission of acridine orange (excitation: 495 nm) was monitored at 530 nm. To determine ion selectivity 5 mM NaAc, LiAc, KAc or TlAc were added to the reaction mixture after the initial sodium efflux reached equilibrium. Addition of substrates for PaNhaP to the reaction buffer results in proton efflux and fluorescence dequenching. Finally, the remaining proton gradient was dissipated by adding 25 mM $(NH_4)_2SO_4$ in all experiments as a control. Electrogenic transport was investigated by addition of 100 nM valinomycin to the reaction buffer. The temperature was kept constant in a water bath during each experiment. Temperature dependence of transport was measured (triplicates) between 20°C and 45°C by correlating the speed of fluorescence quenching in the mid of the curve drop.

## Radioactive $^{22}Na^+$ uptake assays

PaNhaP was reconstituted in 20 mM choline citrate/Tris pH 4–8, 10 mM $(NH_4)_2SO_4$. The reaction mixture contained 20 mM of the same buffer, 10 mM choline chloride, 1 µCi/ml $^{22}Na^+$ and NaCl concentrations ranging from 1 µM to 5 mM. The pH-profile was determined at 200 µM NaCl. For each reaction 2 µl proteoliposomes were diluted in 200 µl reaction buffer to initiate the reaction. The addition of proteoliposomes to the reaction buffer results in $NH_3$ efflux, producing an outward-directed proton gradient (*Dibrov and Taglicht, 1993*). At the time points indicated, 200 µl samples of the reaction mixture were applied to a 0.2 µm millipore nitrocellulose (Millipore, Billerica, MA) filter and washed with 3 ml $^{22}Na^+$-free reaction buffer. Filters were transferred to counting tubes and 4 ml liquid scintillation cocktail (Rotiszint, Germany) was added. All measurements were performed at room temperature and repeated at least three times.

## Crystallization

Prior to crystallization, the buffer for native protein was exchanged in the final concentrating step to 10 mM Tris/HCl pH 7.4, 100 mM NaCl, 0.15 % Cymal-5. Crystallization was performed in 24-well plates in hanging drops at 18°C. SeMet protein was supplemented with 1 % OG and native protein with 1 % n-octyl-β-thio-maltoside (OTM). The protein solutions were mixed 1:1 with reservoir buffer (native protein: 40 mM Na-Citrate pH 4.0, 100 mM NaCl, 28–33 % PEG 350 MME; SeMet protein: 100 mM

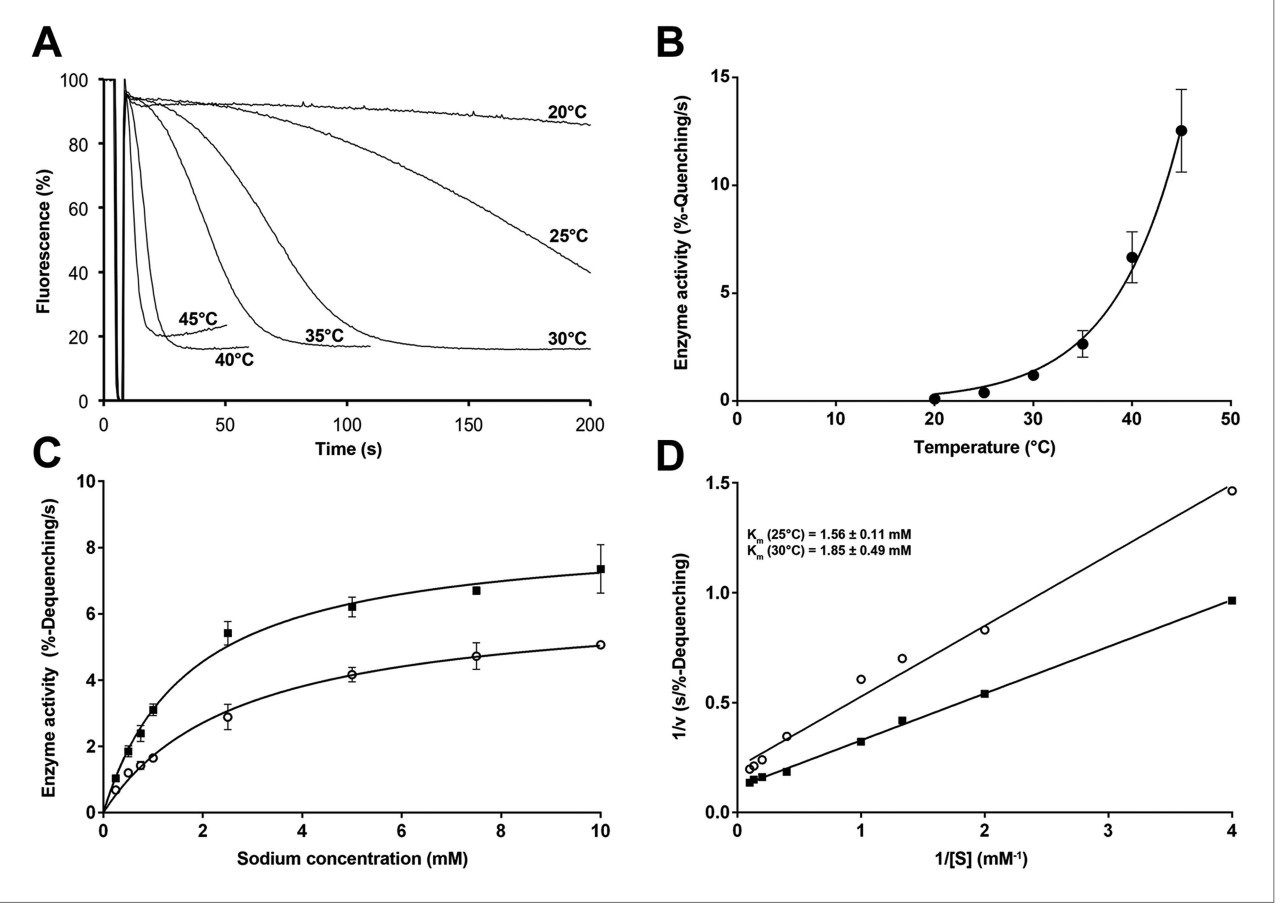

**Figure 7**. Temperature dependence of PaNhaP. (**A**, **B**) At pH 6 transport activity increases by a factor of 2.1 for every 5°C rise in temperature, as measured by sodium efflux under symmetrical pH. The slight rise in fluorescence towards longer times at 40°C and above in A is due to increasing proton leakage of the proteoliposomes. (**C**, **D**) Effect of temperature on substrate affinity at 25°C (empty dots) and 30°C (filled squares) measured by ΔpH-driven sodium uptake in proteoliposomes using Acridine orange fluorescence. In contrast to $v_{max}$, $K_m$ does not change much with increasing temperature (1.56 ± 0.11 mM at 25°C; 1.85 ± 0.49 mM at 30°C).

Tris/HCl pH 8.0, 100 mM $CaCl_2$/$MgCl_2$, 35–40 % PEG 400). Trapezoidal pH 4 crystals grew up to 200 µm within 7 days. At pH 8, long needle-like crystals grew to full size within 3 months. Crystals were vitrified directly in liquid nitrogen for data collection. For thallium soaks, crystals grown at pH 8 were transferred into a buffer containing 100 mM Tris/acetate, 100 mM $MgAc_2$, 40 % PEG 400, 2 mM K-citrate, 0.15 % Cymal-5 and 1 % OG. After five minutes the crystals were transferred to another drop of the same solution containing 25 mM TlAc. Crystals were incubated overnight and vitrified directly in liquid nitrogen.

## Data collection, processing and structure determination

All diffraction data were collected with crystals kept at 100 K at the beamline X10SA of the Swiss Light Source in Villigen, Switzerland. Datasets were processed with XDS (*Kabsch, 1993*) and scaled with AIMLESS in the CCP4 package (*Collaborative Computational Project 4, 1994*). Resolution cut-offs were chosen based on CC1/2 (cross correlation of half datasets), completeness and I/σ(I)-values in high resolution shells (*Karplus and Diederichs, 2012*). Coot (*Emsley and Cowtan, 2004*) was used for model building and the PHENIX package (*Adams et al., 2004*) for refinement. Phases were obtained by single-wavelength anomalous dispersion (SAD) using SeMet crystals. Datasets from two crystals were merged to achieve a high multiplicity and to increase the anomalous signal (*Liu et al., 2011*). The Selenium substructure containing 11 out of 14 possible positions was determined at 5.7 Å using SHELXD (*Sheldrick, 2010*).

Phasing, hand determination, density modification with Parrot (*Zhang et al., 1997*) and initial model building with Buccaneer (*Cowtan, 2006*) was performed with a beta version of CRANK2

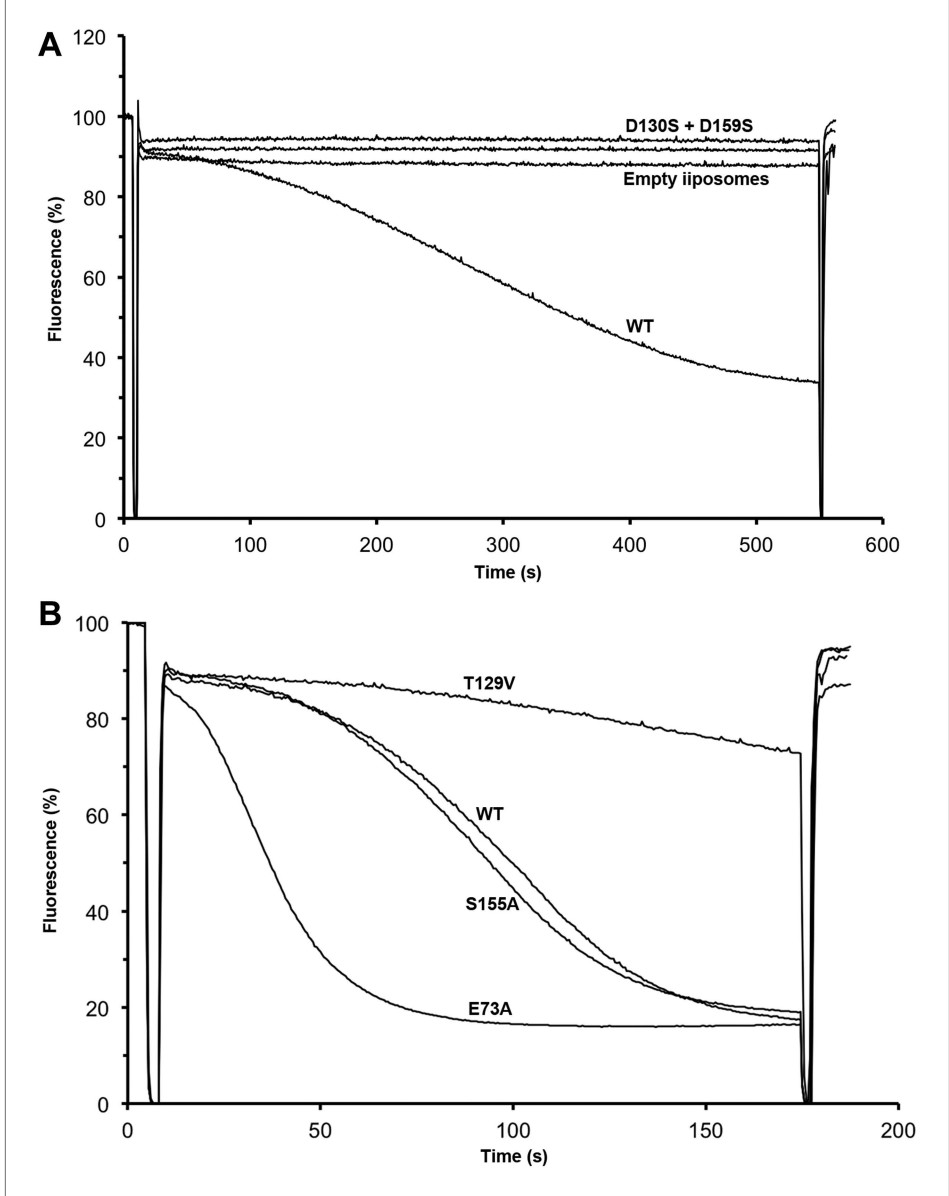

**Figure 8**. Transport activity of binding site mutants. Sodium efflux from proteoliposomes at pH 6 was measured to investigate PaNhaP mutants. Antiport activity establishes a ΔpH across the membrane, which results in acridine orange fluorescence quenching. (**A**) Mutation of Asp130 or Asp159 to serine abolishes transport activity. (**B**) Replacement of Thr129 by valine, as in eukaryotic antiporters, reduces the transport activity. Replacement of Glu73 by alanine increases activity significantly, whereas exchanging Ser155 against alanine has no effect compared to wildtype.

The following figure supplement is available for figure 8:

**Figure supplement 1**. Interface crosslinks.

(*Pannu et al., 2011*). The resulting electron density map was used for manual building of an initial backbone model. Selenium positions were used to assign side chains in initial refinement rounds. Molecular replacement was performed using PHASER (*McCoy, 2007*) with the assigned dimer model to extend the resolution to 3.15 Å. The final pH 8 model was used for molecular replacement to phase the pH 4 structure and the thallium bound structure at pH 8. Superimpositions were performed using secondary structure superimposition (*Krissinel and Henrick, 2004*) within Coot (*Emsley and Cowtan, 2004*).

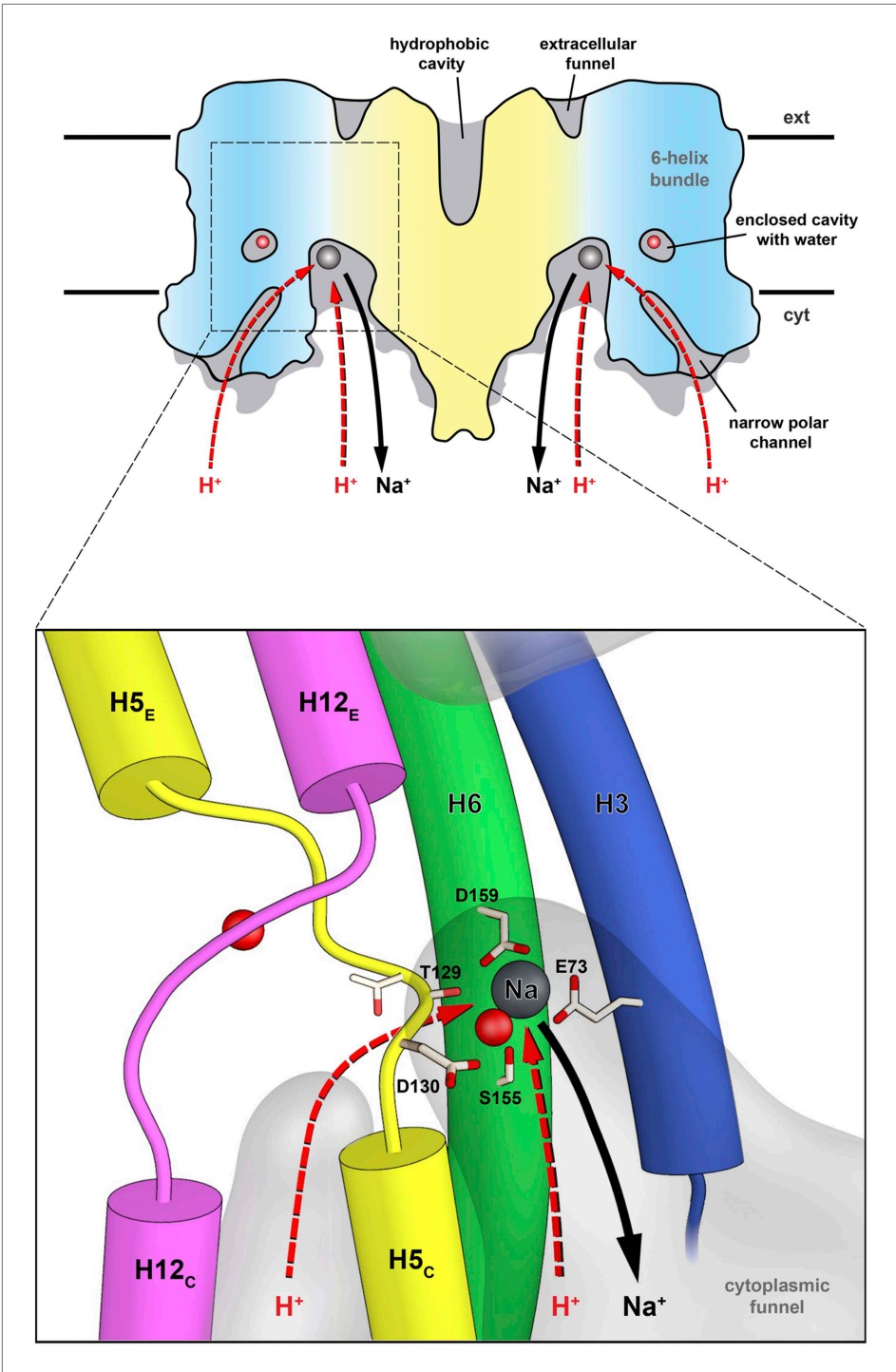

**Figure 9**. Substrate ion exchange on the cytoplasmic side. The substrate-binding site of PaNhaP is located between the unwound stretches in the six-helix-bundle and the interface domain. The substrate ion is bound by acidic sidechains and polar groups in the bundle helices H5 and H6, and a glutamate in the interface helix H3 at the deepest point of the cytoplasmic funnel. While the funnel extends between the six-helix bundle and the dimer interface, the narrow polar channel is defined by the bundle helices $H5_C$, $H12_C$, H6 and H13. Protons may approach the binding site either through the cytoplasmic funnel, or through the narrow polar channel (red arrows). A proton displaces the bound substrate ion, which escapes to the cytoplasm (black arrow). Employing the narrow polar channel as the proton path would separate the $Na^+$ ion and proton currents on the cytoplasmic side, which may be advantageous at high transport rates.

Figures were prepared with PyMOL (*DeLano and Lam, 2005*). The potential surface was calculated with pdb2pqr (*Dolinsky et al., 2007*) and APBS (*Baker et al., 2001*). Analysis of transport pathways, channels and cavities was performed with Hollow (*Ho and Gruswitz, 2008*) and visualized within PyMOL.

## Author information

Atomic coordinates and structure factors have been deposited with the PDB under accession codes: 4cz8 for the pH 8 SeMet structure, 4cz9 for the pH 4 structure and 4cza for the thallium-bound structure at pH 8.

## Acknowledgements

We thank Sabine Häder for technical assistance, Cristina Paulino, Gerhard Hummer, Klaus Fendler and Christine Ziegler for discussion, and Pavol Skubak for the beta version of the Crank2 software. Crystals were screened at the beamlines id23.1 and id29 of the European Synchrotron Radiation Facility (ESRF)/Grenoble and data were collected at the Max Planck beamline PXII of the Swiss Light Source (SLS). This work was funded by the Max Planck Society; the Frankfurt Cluster of Excellence Macromolecular Complexes; the Frankfurt International Max Planck Research School; and SFB 807 'Transport and communication across biological membranes'.

## Additional information

### Competing interests

WK: Reviewing editor, *eLife*. The other authors declare that no competing interests exist.

### Funding

| Funder | Grant reference number | Author |
| --- | --- | --- |
| Max-Planck-Gesellschaft (Max Planck Society) | | David Wöhlert, Werner Kühlbrandt, Özkan Yildiz |
| Deutsche Forschungsgemeinschaft | Cluster of Excellence, Macromolecular Complexes | Werner Kühlbrandt |
| Max-Planck-Gesellschaft (Max Planck Society) | International Max Planck Research School, Frankfurt | David Wöhlert, Werner Kühlbrandt |
| Deutsche Forschungsgemeinschaft | Transport and communication across biological membranes SFP807 | David Wöhlert, Werner Kühlbrandt |

The funders had no role in study design, data collection and interpretation, or the decision to submit the work for publication.

### Author contributions

DW, Purified and crystallized the protein, Performed the functional analyses; WK, Conception and design, Analysis and interpretation of data, Drafting or revising the article; ÖY, Conception and design, Acquisition of data, Analysis and interpretation of data, Drafting or revising the article

## Additional files

### Major datasets

The following datasets were generated:

| Author(s) | Year | Dataset title | Dataset ID and/or URL | Database, license, and accessibility information |
| --- | --- | --- | --- | --- |
| Woehlert D, Kuhlbrandt W, Yildiz O | 2014 | Structure of the sodium proton antiporter PaNhaP from Pyrococcus abyssii at pH 8 | http://www.pdb.org/pdb/search/structidSearch.do?structureId=4CZ8 | Publicly available from RCSB Protein Data Bank. |

| Woehlert D, Kuhlbrandt W, Yildiz O | 2014 | Structure of the sodium proton antiporter PaNhaP from Pyrococcus abyssii at pH 4 | http://www.pdb.org/pdb/search/structidSearch.do?structureId=4CZ9 | Publicly available from RCSB Protein Data Bank. |
|---|---|---|---|---|
| Woehlert D, Kuhlbrandt W, Yildiz O | 2014 | Structure of the sodium proton antiporter PaNhaP from Pyrococcus abyssii with bound thallium ion | http://www.pdb.org/pdb/search/structidSearch.do?structureId=4CZA | Publicly available from RCSB Protein Data Bank. |

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
