## [Decision Letter]

Thank you for sending your work entitled ‘Structure and substrate ion binding in
the sodium/proton antiporter PaNhaP’ for consideration at *eLife*.
Your article has been favorably evaluated by Michael Marletta (Senior editor), Richard
Aldrich (Reviewing editor), and 2 reviewers, one of whom, Rajini Rao, has agreed to
reveal her identity. A further reviewer remains anonymous.

The Reviewing editor and the reviewers discussed their comments before we reached this
decision, and the Reviewing editor has assembled the following comments to help you
prepare a revised submission.

This paper from the Kuhlbrandt group reports a trio of new structures of PaNhaP, an
archael homolog of the NhaA family of Na/H exchangers, including the first structure of
a Na/H exchanger that includes a cation, in this case a Thallium ion. Together, these
structures represent a significant step forward in our understanding of this exchanger
family, especially by visualizing a candidate cation binding site (and it's nice
that Tl actually can drive transport, suggesting that this is indeed
*the* cation binding site). Mutations of residues involved in cation
binding have dramatic effects on transport activity consistent with the proposed role of
the identified site. The crystal structures are very nice and the activity assays seem
to have been carefully performed. The work is clearly presented and well written. The
following comments should be addressed to improve the paper:

1) This manuscript should be combined with the accompanying one as a single revised
submission.

2) Role of changes in the dimer interface. The authors report that the low pH form of
the crystal primarily shows changes at the dimer interface but the actual structural
rearrangements seem quite small. More of a concern though is how to interpret these
changes in the context of mechanism. The state of the transporter in the low pH crystal
is not at all clear-is it still inward facing? Lacking information about the state, we
find it hard to conclude that the changes at the dimer interface ‘relay
allosteric changes from the other protomer’. Indeed, the authors'
interpretation of the structure implies that the low pH form of the protein should have
substantially different Na affinity than the pH 8 form (if indeed they reflect the same
overall state), but this prediction is not tested with the experiments shown here.
Indeed, the Km for Na of 505 uM at pH 5 seems to shift to ∼200 mM at pH 6 but the
structures are at pH 4 and 8, where activity is substantially different. The structures
suggest that actual binding affinities could indeed be measured at pH 4 and 8, which
would be essential to support the authors' interpretation. In addition, we find the
superposition of structures presented in Figure 5—figure supplement 1 to capture the overall comparison of structures
much better than the one in Figure 5 itself and
would include at least one of these in the primary figure.

3) The acridine orange assay used in both papers to measure proton flux is an excellent
assay for qualitative assessment of proton flux. However, the actual mechanism of
acridine orange is unknown in detail and it is impossible to quantitatively measure pH
change with this assay. Therefore the relative rates as a function of pH in Figures 5 and 6 are unreliable and should be
omitted. Na22 flux could be used to measure these rates if desired, or a more
quantitative pH probe, like pyranene.

4) The discussion of ‘Self-regulation of transport activity’ is completely
disconnected from the evidence presented in the paper. If the authors wish to discuss
this, they need to provide some experimental or computational support for their claims.
They discuss ‘pH-dependent affinity’ but show no evidence that the
affinity is indeed pH dependent beyond Kms at pH 5 and 6. Whether these values actually
represent affinity depends on a range of assumptions which may or may not be valid for
this protein.

---

## [Author Response]

1) This manuscript should be combined with the accompanying one as a single revised
submission.

For a number of reasons we prefer to keep the manuscripts separate. One reason is
authorship. The x-ray and electron crystallographic structure determination were two
separate projects done by two PhD students. Merging the manuscripts would mean an
injustice to one of them, and shared first authorship would not reflect the different
contributions correctly. The other, more important reason is content. A merged
manuscript would become unwieldy, unless important information is omitted, which we do
not want to do. We therefore decided to revise the PaNhaP manuscript and to pursue
publication of the MjNhaP1 manuscript separately.

2) Role of changes in the dimer interface. The authors report that the low pH form of
the crystal primarily shows changes at the dimer interface but the actual structural
rearrangements seem quite small.

The structural rearrangements at the PaNhaP interface cannot be described as small.
Video 2 and Figure 5 plus supplements clearly show that sidechains in helix H10
move by up to 8A as the pH changes from 8 to 4. Moreover the changes are not confined to
the interface but propagate through the whole protomer, including the loops connecting
the trans-membrane helices.

More of a concern though is how to interpret these changes in the context of mechanism.
The state of the transporter in the low pH crystal is not at all clear-is it still
inward facing?

PaNhaP is in an inward-open conformation both at pH8 and at pH4. This is now stated
explicitly in the revised manuscript. The substrate-binding site is accessible from the
cytoplasm through the cytoplasmic funnel but not from the extracellular side under both
conditions. However, the narrow side channel that leads from the cytoplasmic surface to
the ion-binding site is blocked at pH4 by side chain rearrangements, as stated in the
revised manuscript.

It is important to note that the pH-dependent allosteric change of the dimer is
different from the inside-open to outside-open transition in the transport cycle of the
protomer. This is now stated explicitly in the revised manuscript. Rather, the
allosteric change increases the affinity of the binding site for the substrate ion from
Km = 500 µM to K0.5 = 25 µM. This extends the range for
high-affinity substrate binding by ∼1 pH unit from acidic towards neutral
conditions, as may be necessary for efficient Na^+^/H^+^
exchange at physiological pH in this particular organism. The pH-dependent binding
affinity will be the subject of a future molecular dynamics study that goes well beyond
the scope of the present manuscript.

Lacking information about the state, we find it hard to conclude that the changes at the
dimer interface ‘relay allosteric changes from the other protomer’.

It is evident from Video 2 in the revised
manuscript (Video 1 in the
original manuscript) that the pH-dependent repulsion of protonated histidines 292 causes
conformational changes, and that this mechanism can only work with two protomers next to
one another in the dimer. Therefore the changes do indeed relay allosteric changes
between protomers. However, we agree that it may be better to say that
‘conformational changes caused by repulsion of the protonated histidines at the
dimer interface are relayed to the ion binding site to modulate the Na^+^
binding affinity in a pH-dependent manner’. This is now stated in the revised
manuscript.

To show the pH-induced differences more clearly, we have added Figures 1 and 2 to Figure 5 as stereo images in the revised manuscript.

Indeed, the authors' interpretation of the structure implies that the low pH form
of the protein should have substantially different Na affinity than the pH 8 form (if
indeed they reflect the same overall state), but this prediction is not tested with the
experiments shown here.

Both structures do indeed show the same overall state (inward-open), as explained above.
Because the antiporter is only minimally active at pH4 and pH8 (see Figure 4), we are unable to measure the binding affinity under
these conditions with the methods available to us. However, the Na binding affinities at
pH5 and pH6 are substantially different, as indicated by the K_0.5_ for the
cooperative antiporter at pH6 that we have now added to Figure 4. Compared to the K_m_ at pH5 in Figure 4, this indicates a roughly 20-fold increase in Na binding
affinity.

In the revised manuscript we have emphasized the probable role of protonation states of
acidic residues in the binding site in pH-dependent activity changes. For example,
Asp130, which is involved directly in substrate-ion coordination, changes its
conformation in response to pH, and this directly affects the coordination geometry.
This should now be clearer in the new Figure 5—figure supplement 2, which shows the superposition of the
ion-coordinating residues in both protomers in stereo.

Indeed, the Km for Na of 505 uM at pH 5 seems to shift to ∼200 mM at pH 6 but the
structures are at pH 4 and 8, where activity is substantially different.

We have no idea how the referees arrive at the conclusion that the K_m_ at pH6
should be 200 mM. There was no K_m_ value or even a K_0.5_ value for
this pH in Figure 4 or anywhere else in the
original manuscript. We have added the K_0.5_ value, which is 25 µM, to
Figure 4 of the revised manuscript.

The structures suggest that actual binding affinities could indeed be measured at pH 4
and 8, which would be essential to support the authors' interpretation.

As explained above, PaNhaP is essentially inactive at pH 4 and 8, so these measurements
are not feasible by the methods available to us.

In addition, we find the superposition of structures presented in Figure 5—figure supplement 1 to capture the overall
comparison of structures much better than the one in Figure 5 itself and would include at least one of these in the primary
figure.

In the revised manuscript, we replaced Figure 5
by Figure 5—figure supplement 1 in the
original manuscript. Video 1 of the original
manuscript is now Video 2, which shows the
pH-induced conformational changes very clearly. Stereo pairs of original Figure 5 are now provided as supplements to this
Figure, making the conformational changes even clearer.

3) The acridine orange assay used in both papers to measure proton flux is an excellent
assay for qualitative assessment of proton flux. However, the actual mechanism of
acridine orange is unknown in detail and it is impossible to quantitatively measure pH
change with this assay. Therefore the relative rates as a function of pH in Figures 5 and 6 are unreliable and should be
omitted.

We do not understand this comment. Figure 5 of
the original manuscript shows pH-induced conformational changes. Figure 6 shows the temperature dependence of transport.

Figure 4–figure supplement 1 (both in the original and the revised manuscript)
does show acridine orange measurements as a function of pH but these measurements are
only qualitative to confirm transport activity under symmetrical pH conditions. All
quantitative measurements of pH-dependent activity were performed with radioactive
^22^Na assays.

Na22 flux could be used to measure these rates if desired, or a more quantitative pH
probe, like pyranene.

This is exactly what we have done, as explained in many different places in the original
manuscript.

4) The discussion of “Self-regulation of transport activity” is completely
disconnected from the evidence presented in the paper. If the authors wish to discuss
this, they need to provide some experimental or computational support for their claims.
They discuss ‘pH-dependent affinity’ but show no evidence that the
affinity is indeed pH dependent beyond Kms at pH 5 and 6. Whether these values actually
represent affinity depends on a range of assumptions which may or may not be valid for
this protein.

Experimental evidence includes the observation that mutation of His292 to cysteine
(Figure 7–figure supplement 1) did not alter the pH dependence, as stated in the
revised manuscript. Additional experimental support is provided by SSM measurements with
MjNhaP1 as cited in the manuscript.